# Recent Progress in Conjugated Polymers-Based Donor–Acceptor Semiconductor Materials for Photocatalytic Hydrogen Evolution from Water Splitting

**Yanhui Zhao** [1]**, Jingfu Sheng** [1]**, Xiaobo Zhao** [1,*]**, Jian Mo** [1]**, Jilong Wang** [2]**, Zhuang Chen** [1]**, Hongjun Dong** [2] **and Chunmei Li** [2,*]

[1] College of Chemistry, Baicheng Normal University, Baicheng 137000, China; zhaoyanhui198523@163.com (Y.Z.); shengjingfu0912@163.com (J.S.); mj82430@foxmail.com (J.M.); cz20020608@foxmail.com (Z.C.)

[2] Institute of Green Chemistry and Chemical Technology, School of Chemistry and Chemical Engineering, Jiangsu University, Zhenjiang 212013, China; jlwang0229@163.com (J.W.); hjdong@ujs.edu.cn (H.D.)

[*] Correspondence: zhaoxiaobo@bcnu.edu.cn (X.Z.); dhjlcm@ujs.edu.cn (C.L.)

**Abstract:** Exploration of high-efficiency stabilization and abundant source-conjugated polymers semiconductor materials with suitable molecular orbital energy levels has always been a hot topic in the field of photocatalytic hydrogen evolution (PHE) from water splitting. In the recent years, constructing the intramolecular donor–acceptor (D–A)-conjugated architecture copolymers has been proved as one of the most excellent photocatalyst modification tactics for optimizing the PHE properties because of unique advantages, including easy regulate band-gap position, fast transfer charge carrier in the intramolecular architecture, superior sunlight absorption capacity and range, large interfacial areas, and so forth. Therefore, in this minireview, we summarize the latest research progress of D–A architecture semiconductor materials for PHE from water splitting. First, we briefly overview the fundamental description and principles for the construction D–A heterostructures in the photocatalytic system. After that, the application of D–A architecture photocatalyst for PHE reaction over different classes of organic semiconductors have been discussed in detail. At last, the present development prospects and future potential challenges of D–A architecture materials are proposed. We hope this minireview has some parameter values for the further developments of intermolecular special structured organic semiconductor material in the future PHE research.

**Keywords:** organic semiconductor; intramolecular donor–acceptor architectures; photocatalysis; hydrogen evolution; water splitting

## 1. Introduction

With the rapid development of industrialization in modern society, the consequent energy shortage issue is becoming more and more serious. Exploitation of the green and pollution-free new energy sources has become the only way which must be passed to the sustainable development of humankind and nature. As the typical alternative and renewable energy source, hydrogen ($H_2$) has drawn on extensive attention due to some unique advantages, including wide range of sources, high combustion calorific value and energy utilization rate, green non-pollution products [1–7]. Among the various reported strategies for producing hydrogen, the photocatalytic $H_2$ evolution (PHE) from water splitting based on endless stream of solar energy has been considered as the optimizing pathway because it can convert solar to hydrogen energy directly without using complex technology and causing secondary pollution [8–16]. As for the photocatalytic technology, the most important step is to select the appropriate semiconductor photocatalyst to satisfy the thermodynamic and dynamic characteristics of solar energy conversion and obtain

the desired solar-to-hydrogen conversion efficiency [17–20]. The currently reported photocatalysts are mainly divided into inorganic semiconductor and organic semiconductor; the latter displays the remarkable photocatalytic activities in contrast to the former owing to many unique advantages such as easily adjustable structure, broad light absorption range, excellent electrical conductivity, forceful bonding effects, favorable stability, and low toxicity [21–23]. However, a single non-metallic photocatalyst cannot meet the needs of practical applications owing to lower PHE activity resulting from the poor charge carrier separation efficiency, solar energy utilization, and stability [24,25]. In recent years, a variety of modification methods have been developed over the non-metallic photocatalyst to boost PHE activity and ease the current energy crisis.

Among the reported modification strategies of nonmetallic photocatalysts so far, intramolecular regulatory manners have been confirmed to play a very positive function in optimizing PHE activity owing to their adjusted electronic band-gap structure, induced charge redistribution, enhanced absorption capacity and widened absorption range for solar light, and facilitated migration and separation of photogenic charge carriers [26–28]. In particular, the construction of intramolecular-conjugated polymers for electron donor–acceptor (D–A) architectures with different electron appetency of building units have been extensively applied in the PHE field, in which the electron D-units and A-units can commendably regulate band-gap position and electrons redistribution resulting from pushing and pulling the highest occupied molecular orbitals (HOMO) and lowest unoccupied molecular orbitals (LUMO), respectively [29–33]. Furthermore, the intramolecular charge carrier transfer from donor units to acceptor units over D–A molecular architecture in the PHE process can expedite Frenkel excitons migration and reduce the Coulomb interaction of photoinduced charge carriers owing to the formation of internal polarized electric field. Additionally, self-organization of D and A units can also possess the large interfacial areas by the formation of the bicontinuous interpenetrating network structure contributing to the transfer of charge carrier. Specifically, some reported organic semiconductors have been used to fabricate D–A polymeric coupling heterojunction, such as g-$C_3N_4$ [34], 1,3,6,8-tetrabromopyrene [35], benzo[b]thiophene-2-carboxylic acid [36], polyarylether [37], and so on. The building units of the above involved several characteristics or requirements: (1) Organic molecules usually possess relatively large dipole in favor of the formation of build-in electric field; (2) it should have unique electron donor or acceptor properties with regard to the transfer and separation of charge carrier; (3) relatively high molar absorption efficiency for enhanced light-harvesting capacity; (4) the π-electron system should be highly delocalized reducing the likelihood of recombination of photoproduced electrons and holes in the photocatalytic reaction process. Based on the above multiple advantages, a growing number of organic D–A coupling systems photocatalyst with various donor and acceptor building units have been established, and displayed great potential in the PHE field with desired transfer and separation capabilities of charge carrier.

Therefore, in this minireview, we have summarized the recent research progress of D–A architectures heterojunction photocatalysts for PHE from water splitting. First, we present a brief and fundamental description of D–A heterostructures to offer some insightful principles of the construction of this photocatalytic system. Second, the possible intramolecular charge transfer and PHE reaction mechanism over D–A heterostructures have been proposed in this review. Afterward, the composition of D–A heterostructures involved in carbon-nitride-based polymers (g-$C_3N_4$), covalent organic frameworks (COFs), covalent triazine-based frameworks (CTFs), and other photocatalysts for PHE reaction are discussed in detail and analysis of the current research results was carried out. Finally, the challenges, opportunities, and development prospects based on non-metallic D–A heterostructures photocatalyst in the PHE domain have been proposed. We sincerely hope this minireview can contribute to the in-depth studies on highly effective D–A heterostructures photocatalysts for meeting the prospective practical applications.

## 2. Fundamental Description and Mechanism of D–A Heterostructures

### 2.1. Fundamental Description of D–A Heterostructures

The space charge region formed by the interface between different P-type and N-type semiconductors, which is called "P-N heterojunction" in physics, is one of the most effective strategies for photoinduced charge separations. Analogous to the concept of P-N heterojunction, D–A heterostructure can be defined as a system in which the electron-donor and electron-acceptor are directly connected through covalent or non-covalent interaction (such as supramolecular assembly), and the donor and acceptor units should be semiconductive components [30,38,39]. Electrons in the donor or acceptor components are photoexcited to produce coupled electron-hole pairs (Frenkel excitons). After migrating to the D–A interface, these excitons dissociate into free charges driven by the energy level difference between the donor unit and acceptor unit. The electrons are driven into the acceptor phase and the holes into the donor phase. Subsequently, the charge carriers are further transported to the corresponding electrodes, respectively, thus forming a long-lived charge-separated state (Figure 1A) [40–45]. The construction of a interpenetrating structure and a bicontinuous model between the donor and the acceptor can increase the contact of the active interfaces and form the so-called bulk heterojunctions (BHJs), thus significantly accelerating the separation of photogenerated charge carriers, while providing a continuous and short channel for charge migration to the electrodes (Figure 1B,C) [30,46].

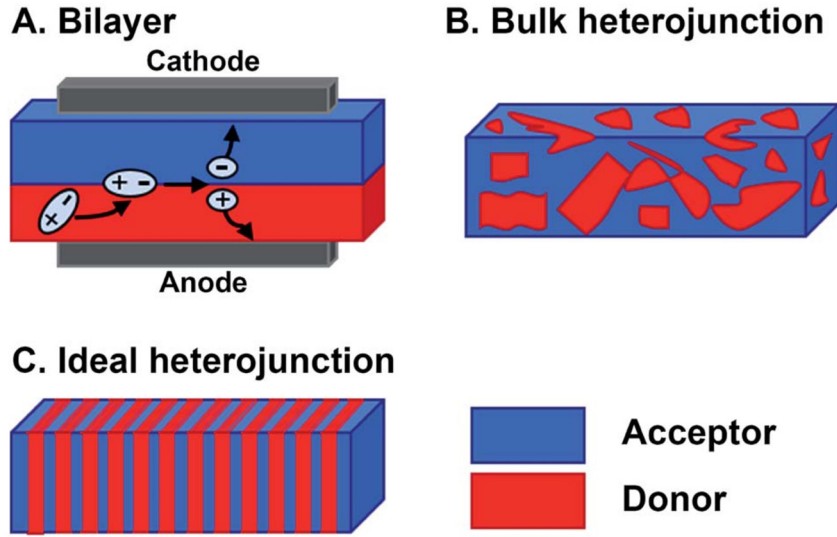

**Figure 1.** Donor–acceptor heterojunction morphologies. (**A**) Bilayer structure. (**B**) BHJ and (**C**) heterojunction optimized to increase junction area, while providing contiguous pathways for charges to migrate to the electrodes [41].

### 2.2. Mechanism of D–A Heterostructures

According to the mechanism of photocatalytic reaction, the PHE process can be improved by three strategies: (1) Enhancing the range and intensity of visible light absorption; (2) promoting the separation and transfer of charge carriers and inhibiting the rapid recombination of electron-hole pairs; (3) improving the redox activity of surface reaction, in which the separation and transfer of electron-hole pairs has been considered to be the most crucial step in the photocatalytic process. The introduction of D–A structure is an efficient method to accelerate exciton dissociation and migration in conjugated polymers. By selecting the different electron-deficient and electron-rich units to form the D–A structures in conjugated polymers can not only adjust the molecular orbital energy level and intermolecular interaction, but also can extend the conjugated system, thus affecting the intramolecular charge transfer in conjugated polymers [47–51].

Intramolecular charge transfer (ICT) describes the transfer of electrons from donor to acceptor or the holes from acceptor to donor during photoexcitation. The strength of ICT

depends on the relative electron affinity of the two units. Exciton binding energy ($E_b$) is a key parameter involved in charge separation in D–A structure, and it can be significantly reduced by appropriate regulation of ICT process. Meanwhile, adjusting the energy levels of the HOMO and LUMO can change the band gap and affect the exciton production. The spontaneous and continuous transfer of charge from electron donor to electron acceptor mainly depends on the energy of photoexcited excitons ($E_{exc}$), the energy of charge transfer (CT) states ($E_{CT}$), and the energy of LUMO level of electron acceptor ($E_{LUMO}$). Obviously, $E_{exc}$ is generally larger than $E_{CT}$ and $E_{LUMO}$. Simultaneously, the charges can continuously transfer only when the $E_{CT}$ is also larger than the $E_{LUMO}$ of the electron acceptor. Otherwise, the excited charges will quickly reassemble back to the ground state. Thus, increasing the energy difference between the donor and the acceptor and minimizing the energy loss in the CT state can facilitate a reduction in $E_b$ and enhance the charge transfer process which is beneficial to produce more electrons at the LUMO level of the acceptor [41,42,52,53]. All the above involved in charge transfer process are shown in Figure 2.

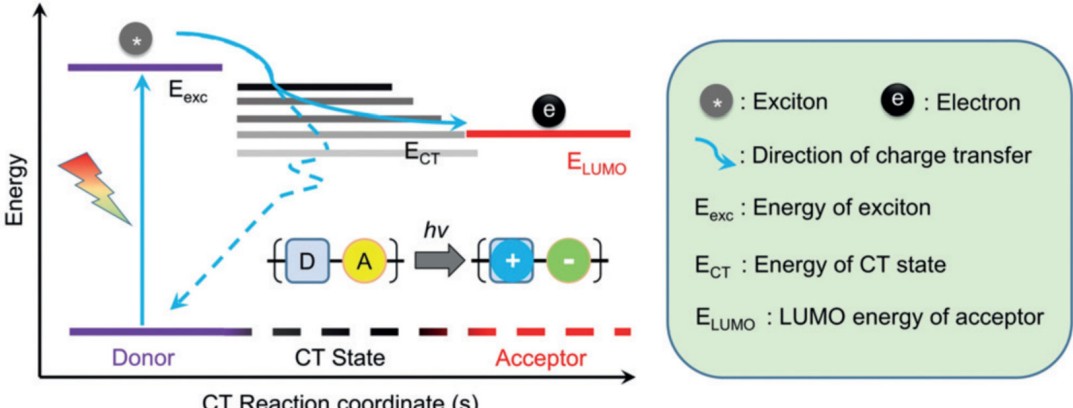

**Figure 2.** Illustration of the energy levels summarizing the main processes involved in charge transfer [52].

In addition, the distance between the donor unit and the acceptor unit is important for charge separation and transfer, which mainly involves the insertion of various aromatic groups in the D–A structure as π-spacers [54]. Different molar ratios of D/A can also significantly affect the separation and transfer efficiency of photogenerated carriers, thus improving the photocatalytic performance [35].

## 3. CPs-Based D–A Photocatalysts for Photocatalytic Hydrogen Evolution

So far, in order to improve the photocatalytic hydrogen evolution performance of conjugated polymers (CPs) and better explore its catalytic mechanism, many material design strategies have been proposed. These improvements include expanding the optical absorption range, adjusting the band gap between the valence and conduction bands, increasing the specific surface area of the catalyst, etc., thus enhancing the mobility of charge carriers and achieving more efficient electron-hole pair separation. D–A-conjugated polymers have proved to be a promising material for the development of high-performance photocatalysts due to their basically meeting the above advantages. Recent studies have focused on conjugated polymers such as g-$C_3N_4$, COFs, CTFs, and others, emphasizing their synthesis methods and structure modification in order to further improve the PHE activity.

### 3.1. D–A Structure Based on g-$C_3N_4$

g-$C_3N_4$, as an efficient and environmentally friendly metal-free photocatalyst, has been widely studied in photocatalysis due to its simple and low-cost synthesis route, suitable optical band gap ($\approx$2.70 eV), high thermal and chemical stability, and good photo-response capability. However, the photocatalytic performance of the pristine g-$C_3N_4$ is seriously

hampered by two aspects: insufficient light absorption in the visible region and rapid recombination of photogenerated carriers. Therefore, researchers have actively developed various methods to enhance the visible light response and accelerate the separation and migration of photogenerated carriers. In recent years, the concept of D–A g-$C_3N_4$ has been introduced into the field of photocatalysis. The synthesis and design of D–A g-$C_3N_4$ photocatalysts have become a research hotspot, which shows the excellent PHE activity. This is because rapid charge transfer channels can be formed between the organic units and g-$C_3N_4$ to realize efficient intramolecular charge transfer, inducing charge redistribution and promoting the separation and transport of the photogenerated charge carriers. Consequently, these factors are conducive to improving the PHE performance of g-$C_3N_4$ [29,55,56]. The design strategy of intramolecular D–A g-$C_3N_4$ is similar to that of molecular-doped g-$C_3N_4$, but the difference is that the doped monomers usually have a strong ability to donate or accept electrons, which can form electron donor–acceptor structures and accelerate the separation of charge carriers. In addition, intramolecular D–A g-$C_3N_4$ is no longer merely substituted by the original fragment or element of g-$C_3N_4$ skeleton, but is prepared by grafting an electron donor or an electron acceptor onto the edge of g-$C_3N_4$ by means of nucleophilic reaction. For instance, our group prepared a novel g-$C_3N_4$ D–A-conjugated polymer with a porous structure by thermal copolymerization of urea and melamine formaldehyde (MF) resin, named as g-$C_3N_4$-$MF_X$. The obtained polymer not only has a large specific surface area, but also enhances visible-light absorption capacity and accelerates charge carriers' separation. As shown in Figure 3a–f, the SEM and TEM images of pure g-$C_3N_4$ present a wrinkled nanosheet-like structure, but when MF resin is copolymerized with urea, the morphology of g-$C_3N_4$-$MF_{100}$ copolymer changed from a nanosheet-like to a porous structure. Compared with pure g-$C_3N_4$, the g-$C_3N_4$-$MF_{100}$ polymer shows clear porous structure. Due to the regular porous structure of g-$C_3N_4$-MFx, it not only increases the specific surface area, exposes more reaction sites, but also shortens the electron transfer distance, greatly improving the PHE activity. As expected, g-$C_3N_4$-$MF_{100}$ exhibits an excellent PHE rate (3612.7 $\mu mol\ h^{-1}\ g^{-1}$), which is over 8.87 times higher than the pure g-$C_3N_4$. However, when MF resin is excessive incorporated into the g-$C_3N_4$ skeleton, the PHE rate decreases sharply. This may be caused by accelerated recombination of charge carriers (Figure 3g,h). In addition, Figure 3i shows an apparent quantum yield (AQY) of 8.6% for g-$C_3N_4$-$MF_{100}$ at 420 nm, which is much higher than that of the most reported g-$C_3N_4$-based D–A-conjugated polymers and porous g-$C_3N_4$. More surprisingly, the PHE rate of g-$C_3N_4$-$MF_{100}$ changes not significantly after six runs within 30 h. It is shown that the porous g-$C_3N_4$-MF has excellent stability (Figure 3j). This work provides a novel strategy for integrating porous structures and intramolecular D–A-conjugated structures into g-$C_3N_4$, significantly enhancing the PHE activity [57].

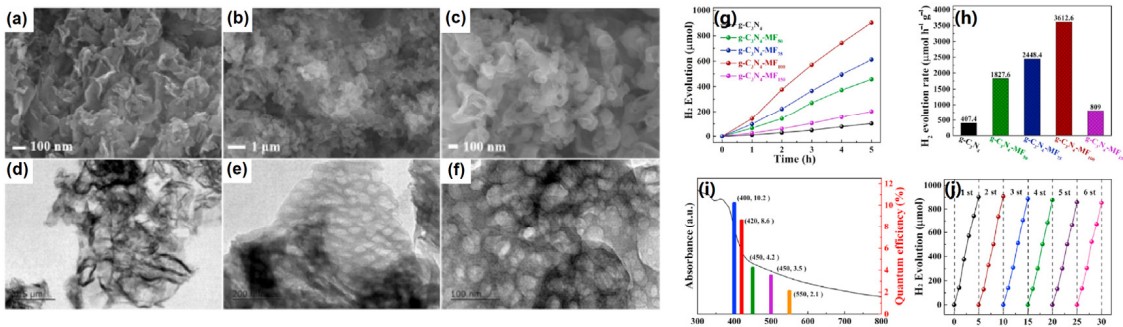

**Figure 3.** Typical SEM images of pure g-$C_3N_4$ (**a**) and the porous g-$C_3N_4$-$MF_{100}$ D–A-conjugated polymer (**b**,**c**). Typical TEM images of pure g-$C_3N_4$ (**d**) and the g-$C_3N_4$-$MF_{100}$ D–A-conjugated polymer (**e**,**f**). (**g**) Time courses of the PHE activity and (**h**) PHE rate for pure g-$C_3N_4$ and g-$C_3N_4$-$MF_x$ under visible-light irradiation. (**i**) The wavelength-dependent AQY for the $H_2$ evolution of g-$C_3N_4$-$MF_{100}$. (**j**) The stability test of PHE over g-$C_3N_4$-$MF_{100}$ [57].

Moreover, in recent work, Che et al. prepared the ultrathin g-$C_3N_4$ (UCN)-based D–A-conjugated copolymers with high degree of crystallinity by copolymerizing benzoyl isothiocyanate (BI) with urea, UCN-$BI_x$. The resultant UCN-$BI_x$ D–A-conjugated copolymers simultaneously achieve excellent PHE activity. As shown in Figure 4a,b, the pure g-$C_3N_4$ and UCN-$BI_{400}$ D–A-conjugated copolymer exhibits an obvious ultrathin nanosheet structure. The ultrathin nanosheets of UCN-$BI_{400}$ D–A-conjugated copolymer become smaller relative to the g-$C_3N_4$. More importantly, the UCN-$BI_{400}$ D–A-conjugated copolymer shows clear lattice fringes, indicating that the UCN-$BI_{400}$ D–A-conjugated copolymer has good crystallinity (Figure 4c). The improved crystallinity accelerates the transfer of electrons in the plane, which further enhances the photocatalytic activity. In addition, the element mapping images show the homogeneous distribution of C, N, and O elements in the UCN-$BI_{400}$ D–A-conjugated copolymer (Figure 4d). It is worth noting that the free Gibbs energies of $H_2$ adsorption ($\Delta G_{H*}$) for the ultrathin D–A conjugated copolymer is investigated by density functional theory (DFT) calculations, and the absolute value of the $\Delta G_{H*}$ of UCN-$BI_x$ D–A-conjugated copolymer decreases dramatically (0.05 eV) compared with that of pure g-$C_3N_4$ (0.34 eV). The smaller absolute value of $\Delta G_{H*}$ with UCN-$BI_x$ indicates a more likely PHE reaction. With the addition of BI precursors, the PHE reaction rate of UCN-$BI_x$ D–A-conjugated copolymer increases gradually. The UCN-$BI_{400}$ D–A-conjugated copolymer shows the highest PHE reaction rate, which was 5442.74 $\mu mol\ g^{-1}\ h^{-1}$, approximately 12 times higher than that of the pristine g-$C_3N_4$ (Figure 4e,f). More unexpectedly, the AQY of UCN-$BI_{400}$ D–A-conjugated copolymer reaches 23.3% at 420 nm and 7.0% at 450 nm, respectively, but decreases significantly with the increase in light wavelength (Figure 4g). Most importantly, cyclic experiments show that the PHE activity of UCN-$BI_{400}$ D–A-conjugated copolymer displays no obvious difference after eight runs within 40 h (Figure 4h), revealing the excellent stability of UCN-$BI_x$ D–A-conjugated copolymer. This work presents a novel design concept that effectively combines ultra-thin structures with D–A-conjugated structures, while improving PHE activity and AQY [58].

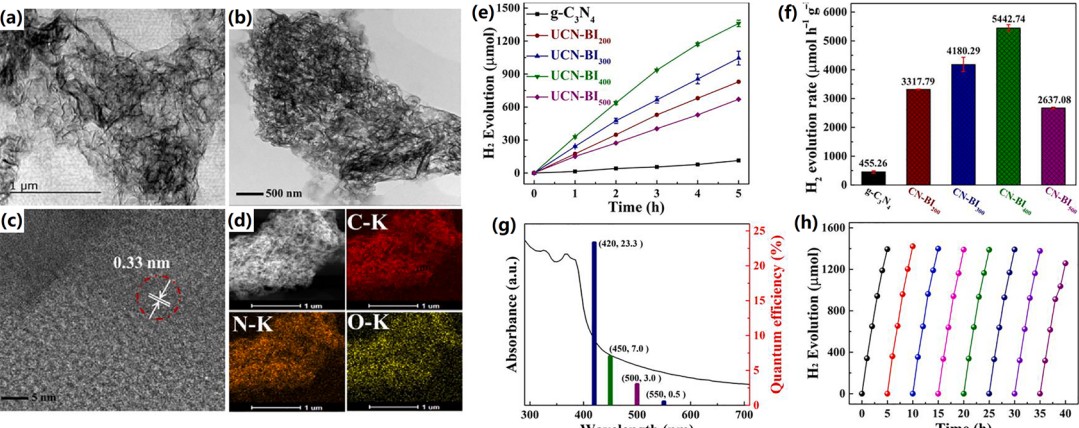

**Figure 4.** Typical TEM images of (**a**) the g-$C_3N_4$ and (**b**) UCN-$BI_{400}$ D–A-conjugated copolymer. (**c**) The HRTEM and (**d**) elemental mapping images of UCN-$BI_{400}$ D–A-conjugated copolymer. (**e**) Time courses of PHE activity and (**f**) PHE rate of the pure g-$C_3N_4$ and UCN-$BI_x$ D–A-conjugated copolymers. (**g**) The AQY and (**h**) stability test of the UCN-$BI_{400}$ D–A-conjugated copolymer [58].

Furthermore, it is universally acknowledged that extended π–π-conjugated electron systems can effectively enhance electron delocalization and reduce photogenerated electron–hole pairs recombination, thus improving the photocatalytic activity. Inspired by the extended conjugation effect in D–A system, Sun and colleagues designed a novel D-π-A mesoporous carbon nitride photocatalyst (J-CNx) by nucleophile substitution reaction between urea and aromatic halogenides as well as Schiff-base reaction between urea and aromatic aldehydes. Inserted benzene as π-spacer by forming covalent bonds C-N (D unit) and C=N (A unit) interrupts the 3s-triazine units while it extends the π–π-conjugated elec-

tron system. Polarization-induced intramolecular charge transfer provides forces-directed migration of electrons, which accelerates the separation and migration of photogenerated charge carriers (Figure 5a). Owing to the large specific surface area in mesoporous structure and significantly enhanced optical and electronic properties, the PHE reaction rate of the optimal sample J-CN20 reaches 2880.98 $\mu mol\ g^{-1}\ h^{-1}$, which is about five times higher than that of the pristine $g-C_3N_4$ (568.45 $\mu mol\ g^{-1}\ h^{-1}$) (Figure 5b). Furthermore, the AQY of J-CN20 achieves 5.49% at 420 nm, but remains 1.87% at 500 nm (Figure 5c). In addition, as shown in Figure 5d, the PHE performance of J-CN20 retains at a high level after five cycles of testing, indicating that J-CN20 has robust stability and excellent resistance to photocorrosion (Figure 5d). Finally, other aromatic heterocyclic compounds (i.e., furan, pyridine, and thiophene) were investigated as π-spacer to construct D-π-A structures, in which the thiophenyl D-π-A structure exhibits the best PHE performance (3882.09 $\mu mol\ g^{-1}\ h^{-1}$), which is about seven times greater than that of $g-C_3N_4$ (Figure 5e). This work provides an inspiration for the design of D–A polymers with extended conjugation effect to achieve efficient solar energy conversion [56].

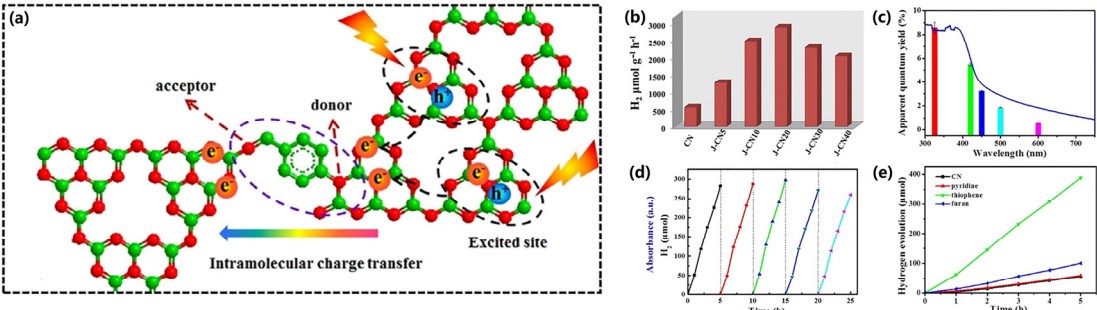

**Figure 5.** (**a**) Schematic for intramolecular charge transfer in D-π-A type conjugated carbon nitride. (**b**) PHE rates for CN and all J-CN samples under visible light irradiation. (**c**) Wavelength-dependent AQY and (**d**) cycling stability test of J-CN20. (**e**) The corresponding PHE rates under visible light irradiation [56].

All the above results explained that $g-C_3N_4$ can be used as the perfect ingredient of D–A architectonic photocatalyst, more suitable synthesis strategies can be used in the future for obtaining the ideal PHE performance. The summary of the PHE activities of some important D–A architectonic based on $g-C_3N_4$ is shown in Table 1 [59–68].

**Table 1.** The summary of PHE activities of some important D–A architectonic based on $g-C_3N_4$.

| Photocatalyst | Light Source | λ | Mass[g]/(Solution)/ Volume [mL] | H$_2$ Evolution Activity (mmol g$^{-1}$ h$^{-1}$) | Quantum Efficiency (%) | Ref. |
|---|---|---|---|---|---|---|
| T-CN-0.010 | 300 W Xe-lamp | λ > 420 nm | 0.05/10% TEOA/100 mL | 1.618 | 4.8% at 420 nm | [59] |
| UCCN | CEL-HXF300 | λ > 420 nm | 0.025/10% TEOA/100 mL | 9.7 | 73.6% at 420 nm | [60] |
| MSCN | 300 W Xe-lamp | λ > 420 nm | 0.05/10% TEOA/100 mL | 1.0856 | - | [61] |
| CDM | PLS-SEX300D | λ > 420 nm | 0.001/10% TEOA/100 mL | 9.454 | 8.41% at 420 nm | [62] |
| NP-CN$_x$ | 300 W Xe-lamp | λ > 420 nm | 0.05/20% TEOA/100 mL | 2.7914 | 6.4% at 400 nm | [63] |
| 3D CN | 300 W Xe-lamp | λ > 420 nm | 0.05/15% TEOA/100 mL | 2.5212 | 8.21% at 420 nm | [64] |
| UCN-BTD | 300 W Xe-lamp | λ > 420 nm | 0.05/10% TEOA/100 mL | 2.44 | 6.8% at 450 nm | [65] |
| CN-abIM$_{0.03}$ | 300 W Xe-lamp | λ > 420 nm | 0.02/10% TEOA/80 mL | 2.566 | - | [66] |
| CN-NaK | LED | λ > 420 nm | 0.05/10% TEOA/38 mL | 11.72 | 60% at 420 nm | [67] |
| CNSO-20 | 300 W Xe-lamp | λ > 420 nm | 0.05/14.3% TEOA/70 mL | 5.02 | 10.16% at 420 nm | [68] |

### 3.2. Covalent Organic Frameworks (COFs)

COFs, as organic polymers with periodic arrangement, are usually formed by condensation of monomers with specific geometric structure, which usually have high crystallinity and high porosity. COFs consist of selected organic building blocks via covalent bond linkages such as imine [69], imide [70], boronic-ester [71], triazine [72], hydrazone [73], and C = C [74] linkages. They exhibit high thermal and chemical stability due to robust covalent bond connections between the structural units relying on the size of building blocks; two-dimensional (2D) or three-dimensional (3D) COFs can be constructed. In addition, various structural units of COFs are highly planar and conjugated, such as triazine, pyrene, thiophene, porphyrin, etc. These structural units form regular stacking and π-delocalization system in the COFs framework, so that the band structure and electronic properties of COFs can be adjusted reasonably, so that COFs has excellent photoelectric performance. Due to the strong π-force among the donor units and the acceptor units, the ordered arrangement of D and A fragments in donor–acceptor COFs (D–A COFs) can effectively improve the separation efficiency of photogenerated electron-hole pairs and accelerate the directed charge transfer. Combined with the above advantages, COFs are attractive photocatalysts with broad prospects for solving outstanding problems in the field of photocatalysis [75–78].

Many D–A COFs have been found to be effective catalysts for PHE due to their porous network structure, narrow band gap, and extended delocalization in π-conjugated systems. In recent work, Zhang et al. designed and synthesized a series of D–A COFs (Nankai University COFs, NKCOFs) with high crystallinity, porosity, and good stability, in which pyrene was used as the electron donor, and benzothiadiazole and its fluorinated derivatives were selected as electron acceptors. The electronegative fluorine group can effectively enhance the reducing capacity of the electron acceptor, among which the monofluorinated benzothiadiazoles in NKCOF-108 can not only improve the strength of the electron acceptor, but also enable the transfer of electrons to the active centers as much as possible. Notably, the best NKCOF-108 sample exhibits PHE activity at 120 μmol h$^{-1}$, while AQY at 520 nm is 2.96%. The high hydrogen evolution rate of NKCOF-108 is related to its highly ordered layered structure and wide visible light response range. Its high crystallinity and large surface area also ensure that sufficient active sites are available for H$_2$ production [79].

Structural modulation of organic semiconductor building blocks through molecular engineering has proved to be a potentially effective method to improve the carrier mobilities and photoelectrical conversion efficiencies in organic semiconductors, thus further improving PHE performance. For example, Chen and coworkers constructed a series of efficient photocatalysts via the polycondensation of terphenyl based diamines (XTP-BT-NH$_2$) with 4,4′,4″,4‴-(Pyrene 1,3,6, 8-tetraethyl) tetrabenzaldehyde (Py-CHO) under solvothermal conditions named as Py-XTP-BT-COFs (X = H, F, Cl). Herein, the chemical structure of Py-HTP-BT-COF is slightly modulated by chlorination and fluorination strategies (Figure 6a). The results show that both the PHE rate of Py-ClTP-BT-COF and Py-FTP-BT-COF are much higher than that of Py-XTP-BT-COF before and after the addition of Pt cocatalyst. The PHE rate of Py-ClTP-BT-COF (177.50 μmol h$^{-1}$) is 8.2 times higher than that of Py-HTP-BT-COF after the addition of Pt cocatalyst, and the PHE activity is significantly higher than that of most COF-based photocatalysts (Figure 6b,c). In addition, AQY is measured at incident wavelengths of 420, 500, 550, 600, and 650 nm, respectively (Figure 6d), in which the AQE of Py-ClTP-BT-COF reaches 8.45% at 420 nm. Meanwhile, as shown in Figure 6e, no significant decrease is observed in photocatalytic performances of all Py-XTP-BT-COFs during cyclic tests, showing excellent stability. The halogenation strategy of benzothiadiazole moiety not only promotes the effective charge separation, but also significantly reduces the energy barrier for the hydrogen evolution reaction, and obviously improve the photocatalytic performance of COF-based photocatalyst. This work provides a strategy for molecular engineering to regulate novel COFs for future solar-chemical energy conversion [80].

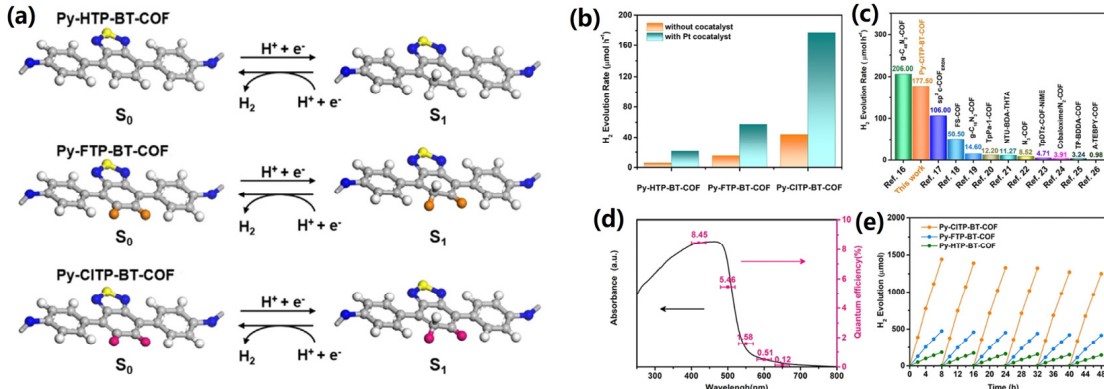

**Figure 6.** (**a**) The proposed PHE reaction pathway on the halogen substituted carbon of Py-XTP-BT-COFs. (**b**) The PHE performances of Py-XTP-BT-COFs with and without co-catalyst. (**c**) The PHE performance comparison of Py-ClTP-BT-COF with other representative COF-based photocatalysts. (**d**) Wavelength-dependent AQY of photocatalytic water splitting by Py-ClTP-BT-COF photocatalyst. (**e**) Time course of H$_2$ evolutions of Py-XTP-BT-COFs [80].

In order to optimize the photocatalytic performance of COFs, the optimal combination between organic building blocks is usually considered to form a suitable COF skeleton. However, covalent linkages between these groups are also important for photocatalytic properties, such as imine COFs, which have been widely studied recently. For example, Yang et al. synthesized three D–A type imine COFs with alternating network structure and high crystallinity by using tris(4-formylphenyl)amine (Tfa), 2,4,6-tris(4-aminophenyl)triazine (Tta), 1,3,5-tris(4-formylphenyl)benzene(Tpa-CHO), and 1,3,5-tris(4-aminophenyl)benzene (TpaNH$_2$) as building blocks. As shown in Figure 7a,b, the skeleton structure features of TtaTfa are observed by high-resolution transmission electron microscopy (HRTEM), and the honeycomb-type pore structure with periodic arrangement can be clearly visualized. Simultaneously, when ascorbic acid is used as the sacrifice electron donor, protonation of imide COFs occurred. Protonated COFs shows significant photocatalytic activity, and the COF TtaTfa which combines the strongest donor (trianiline) and the strongest acceptor (triazine) shows the best PHE performance, achieving a rate of 20.7 mmol g$^{-1}$ h$^{-1}$. The COFs TpaTfa with strong donor and weak acceptor or TtaTpa with weak donor and strong acceptor still have high HER rates of 14.9 mmol g$^{-1}$ h$^{-1}$ and 10.8 mmol g$^{-1}$ h$^{-1}$, respectively (Figure 7c,d). However, when using triethanolamine (TEOA) as the sacrifice electron donor, the activity of the three COFs becomes negligible (Figure 7e). In addition, the results of long-term photocatalytic experiments show that TtaTfa can perform PHE experiment for at least 26 h (Figure 7f). The photocatalytic activity of these COFs indicates that protonation of imine bonds not only improves the charge separation efficiency and light absorption capacity, but also leads to an increase in hydrophilicity. In this work, a novel protonation strategy was proposed to significantly affect the photocatalytic performance of imine COFs [81].

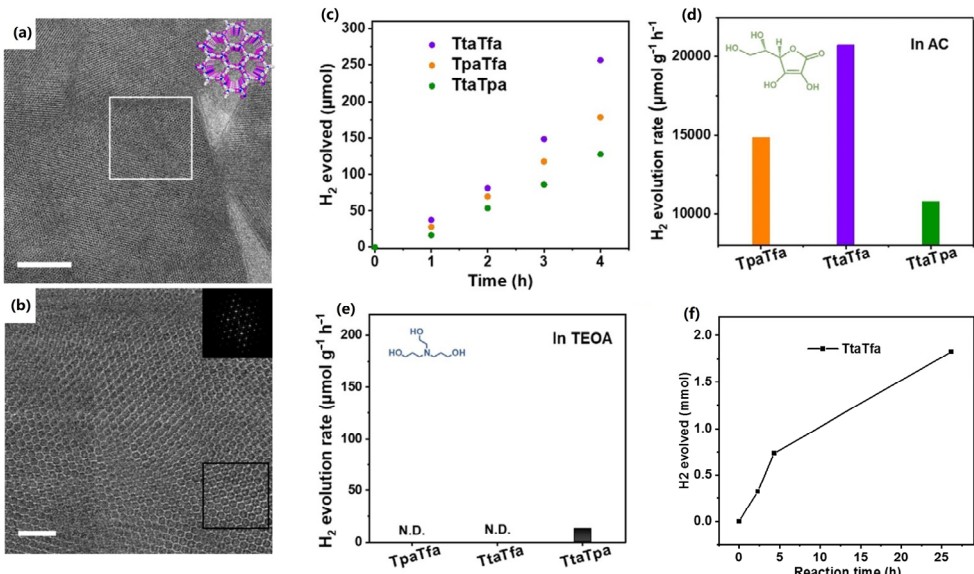

**Figure 7.** (**a**) Low-dose TEM image of TtaTfa under cryogenic conditions. (**b**) Low-dose HRTEM image under cryogenic conditions of the region indicated by white square in (**a**). (**c**) Time course of PHE for TtaTfa, TpaTfa, and TtaTpa. Comparison of photocatalytic PHE rates of the above COFs using AC as SED (**d**) and using TEOA as SED (**e**). (**f**) Time course of PHE for TtaTfa [81].

Although many COFs with excellent PHE rates have been reported, structural optimization of COFs at the molecular and nanoscale levels is helpful to further promote efficient hydrogen evolution. For instance, Yu and coworkers designed and synthesized a D–A type COF with layered pore structure by hydrothermal method using electron-rich tetraphenylethylene (4PE) and electron-deficient thiazol [5, 4-d] thiazole (TZ) as raw materials, named as PETZ-COF. A non-D–A type COF(PEBP-COF) was also synthesized as a control. The nitrogen adsorption isotherm at 77k and BET results reveal that the PETZ-COF shows a high specific surface area of 830 $m^2$ $g^{-1}$ and uniform pore size distribution with two centers at 1.4 nm and 3.5 nm (Figure 8a). In addition, the microscopic morphology of PETZ-COF is observed by using a scanning electron microscope (SEM), and the PETZ-COF shows a hollow spherical structure with diameters between 200 and 400 nm (Figure 8b). TEM also reveals an inerratic pore structure with an aperture of 3.8 nm and a d-spacing of 0.44 nm, corresponding to the (001) crystal surface (Figure 8c,d). Due to its special microstructure, PETZ-COF exhibits remarkable PHE performance in visible light with a rate of 7324.3 μmol $g^{-1}$ $h^{-1}$, an order of magnitude higher than that of PEBP-COF without a D–A structure (217.1 μmol $g^{-1}$ $h^{-1}$) (Figure 8e). Notably, the AQY of PETZ-COF is prominent at 520 nm, indicating high light utilization at this wavelength (Figure 8f). Simultaneously, the hydrogen evolution cycle test is carried out under optimal conditions. The PHE rate of PETZ-COF shows no obvious decrease during the test for more than 20 h, which proves its relative stability and continuous photocatalytic activity (Figure 8g). In addition, although PETZ-COF has a suitable band structure for $O_2$ generation, the oxygen evolution rate (OER) of PETZ-COF is still very small, and it is necessary to continue to explore other necessary thermodynamic conditions to improve the process in subsequent work (Figure 8h). This work provided an effective design and synthesis strategy for highly active COF-based catalysts that are regulated at the molecular and nanoscale levels [82]. In PHE application, COF-based D–A structure photocatalysts may present valuable insight into the high PHE activity in the future. However, efforts are required in developing more efficient patterns to ease the current energy crisis. The summary of PHE performance based on COF-based D–A structure is shown in Table 2 [83–91].

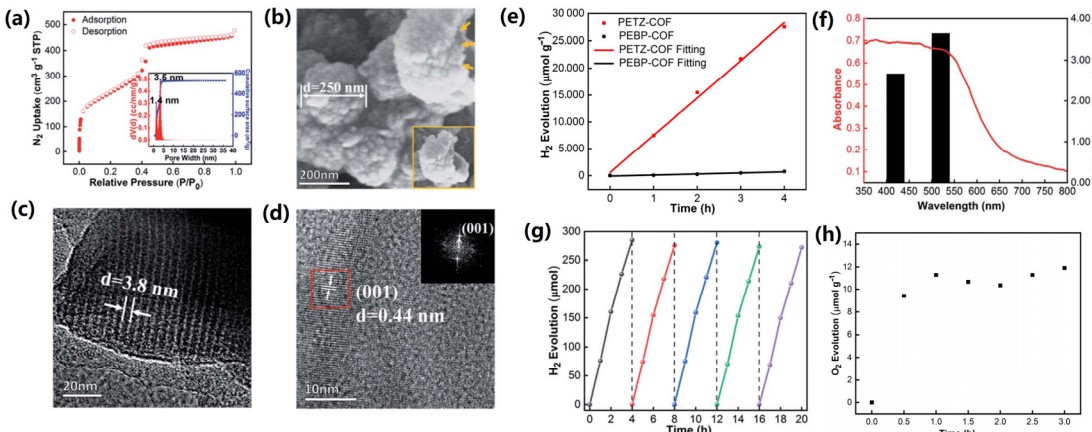

**Figure 8.** (**a**) Nitrogen adsorption isotherm of the PETZ–COF, and the inset shows the derived pore size distribution. (**b**) SEM image of the PETZ–COF. (**c**,**d**) HRTEM images of the PETZ–COF. The inset of the Fourier transform (**d**). PHE rate of the PETZ-COF and PEBP-COF (**e**). AQY of the PETZ–COF at the incident light wavelengths of 420 nm and 520 nm (**f**). (**g**) Cycling stability of $H_2$ evolution of the PETZ–COF tested (**g**). $O_2$ evolution rate of PETZ-COF in pure water (**h**) [82].

**Table 2.** PHE activities of various COFs-based D–A polymers.

| Photocatalyst | Light Source | $\lambda$ | Mass[g]/(Solution)/ Volume [mL] | $H_2$ Evolution Activity (mmol $g^{-1}$ $h^{-1}$) | Quantum Efficiency (%) | Ref. |
|---|---|---|---|---|---|---|
| DABT-Py-COF | 300 W Xe-lamp | $\lambda > 420$ nm | 0.005/0.1 M Aa/10 mL | 5.458 | - | [83] |
| HBT-COF | 300 W Xe-lamp | $\lambda > 420$ nm | 0.005/0.1 M Aa/20 mL | 0.019 | - | [84] |
| Tz-COF-3 | 300 W Xe-lamp | - | 0.05/10% TEOA/100 mL | 43.2 | 6.9% at 420 nm | [85] |
| PyTz-COF | 300 W Xe-lamp | - | 0.01/1 M Aa/20 mL | 2.0724 | - | [86] |
| BTH-3 | 300 W Xe-lamp | $\lambda > 420$ nm | 0.005/0.1 M Aa/- | 15.1 | 1.25% at 500 nm | [87] |
| Zn-Por-TT COF | 300 W Xe-lamp | $\lambda > 420$ nm | 0.005/1 M Aal/50 mL | 8.2 | - | [88] |
| BDF-TAPT-COF | - | $\lambda > 420$ nm | 0.01/1 M Aa/20 mL | 1.39 | 7.8% at 420 nm | [89] |
| CYANO-COF | 300 W Xe-lamp | $\lambda > 420$ nm | 0.02/0.1 M Aa/30 mL | 60.85 | 82.6% at 450 nm | [90] |
| PyTP-COF | 300 W Xe-lamp | $\lambda > 420$ nm | 0.02/0.1 M Aa/30 mL | 7.542 | 0.56% at 420 nm | [91] |

### 3.3. Covalent Triazine-Based Frameworks (CTFs)

The basic skeleton of covalent triazine-based frameworks (CTFs) is composed of alternating triazines and phenyl groups [92]. Compared with other organic semiconductor photocatalysts, CTFs have a larger continuous π-conjugated structure, which enables rapid charge transfer. Meanwhile, the nitrogen atom of the triazine ring contains a lone pair of electrons, which can be easily excited to form high density photoelectrons. In addition, the energy band structure of CTFs can be adjusted by changing the copolymer, by modification, and other methods to increase its charge transfer and separation ability, and improve its chemical and thermal stability. However, the synthesis of CTFs requires harsh conditions such as high temperature and strong acid catalysis, thus easily breaking the regular molecular skeleton. Therefore, developing structurally stable and efficient CTFs remains challenging [93–97].

The adjustment of the ratio and structure of donor-acceptor (D–A) components in organic semiconductors has a significant effect on the efficiency of charge transfer in CTFs photocatalyst. Cai et al. designed and synthesized three D–A type covalent triazine framework materials (BDT-CTFs) using benzodithiophene (BDT) as the electron donor and the triazine ring as the electron acceptor. Because the BDT moieties matches the HOMO and LUMO levels of the triazine units, the direct connection between them not only improves the stability of BDT-CTFs, but also ensures more efficient transfer and

separation of photogenerated charge. In addition, the change in D/A ratio in BDT-CTFs greatly modulated energy band structure and pore structure of BDT-CTFs. Among the three synthesized photocatalysts, BDT-CTF-1 has the highest performance and its PHE rate is up to 4500 μmol h$^{-1}$ g$^{-1}$. The low activities of BDT-CTF-2 and BDT-CTF-3 is attributed to the narrow optical band gap, unfavorable valence band position and pore structure caused by inappropriate D/A ratio, which affects the separation and migration efficiency of electron−hole pairs. This work reveals the intrinsic relationship between the design of D–A structure and the charge transfer, providing a strategy for developing the CTFs with high performance [98].

Organic heterostructures can be reasonably prepared by selectively incorporating different donor units or acceptor units into the covalent triazine frameworks (CTFs). Huang and coworkers reported a novel molecular heterostructure based on covalent triazine frameworks (CTFs). Electron-withdrawing benzothiadiazole (BT) and electron-donating thiophene (Th) groups are selectively added to CTFs by adopting a sequential polymerization strategy, and the resulting heterostructures (CTF-BT/Th) exhibit significantly improved charge-carrier separation efficiency and PHE performance. Meanwhile, the excellent PHE performance of CTF-BT/Th is also attributed to the covalently interconnected molecular heterostructures (Figure 9a). As shown in Figure 9b,c, CTF-BT/Th formed by appropriate hybridization shows obviously enhanced PHE activity, and the best sample CTF-BT/Th-1 reaches the PHE rate of 6.6 mmol g$^{-1}$ h$^{-1}$, which is about 4–6 times greater than those of CTF-BT (1.8 mmol g$^{-1}$ h$^{-1}$) and CTF-Th (1.1 mmol g$^{-1}$ h$^{-1}$), respectively. In addition, the AQY of CTF-BT/Th-1 reaches an astonishing 7.3% at 420 nm (Figure 9d). Besides its excellent photoresponse ability, CTF-BT/Th-1 shows impressive photostability in the cycle test, with only a slight decrease in activity observed after four cycles, which is most likely caused by heavy consumption of TEOA (Figure 9e). This work provides a new strategy for the rational design and preparation of novel molecular heterostructure photocatalysts [99].

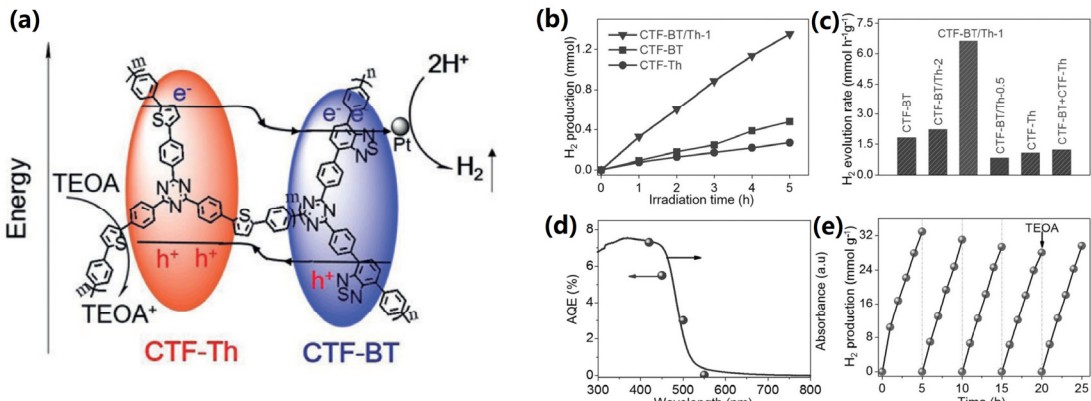

**Figure 9.** (**a**) Illustration of the facilitated charge-carrier separation across the covalently interconnected molecular heterostructure. PHE performance of CTFs incorporating BT and/or Th. (**b**) Time-dependent PHE on CTF-BT/Th-1, CTF-BT, and CTF-Th. (**c**) Average PHE rates on different materials. (**d**) Wavelength-dependent AQY of CTF-BT/Th-1 superimposed with its absorption curve. (**e**) Photostability test of CTF-BT/Th-1 [99].

Recent studies have found that the separation and migration of anisotropic carriers can be achieved by modifying polymer semiconductors by the reasonable incorporation of organic monomers with different structures and properties. For example, Lan et al. synthesized two CTFs with anisotropic molecular structure (CTF-0.5Th and CTF-0.5BT, which are prepared by adding a certain amount (0.5 wt%) of 2,5-dicyanothiophene and benzo[c][1,2,5]thiadiazole-4,7-dicarbonitrile as doping monomers, respectively) by substituting the original phenyl sequence in CTF-B with thiophene (Th) or benzothiadiazole (BT) unit. Theoretically, charge separation and transfer in the pristine CTFs is isotropic because of the regular and uniform distribution of D–A units in the CTFs skeleton (Figure 10a,

top). However, in order to further promote the PHE activity of CTFs in visible light, it is particularly important to develop anisotropic carrier migration (Figure 10a, bottom). In the presence of TEOA as electron donor and Pt cocatalyst, the PHE rate of pure CTF-B under visible light irradiation is 45 µmol h$^{-1}$. Both CTF-0.5Th (62 µmol h$^{-1}$) and CTF-0.5 BT (112 µmol h$^{-1}$) show better photocatalytic activity compared to the pristine CTF-B, suggesting that Th- and BT-modified CTFs have faster carrier separation and transfer and enhanced optical absorption (Figure 10b). Simultaneously, the AQY of CTF-0.5BT reaches 4% at 420 nm and the AQY matches well with the wavelength (Figure 10c). Notably, compared to the bulk structure, CTFs designed with hollow porous morphology can expose more surface-active sites, thus significantly improving the photocatalytic activity of CTFs (Figure 10d). In addition, the photocatalytic activity of CTF-0.5BT remains stable after four consecutive cycles (Figure 10e). This study provides a promising strategy for the design of conjugate polymer photocatalysts with anisotropic carrier transfer effect [100].

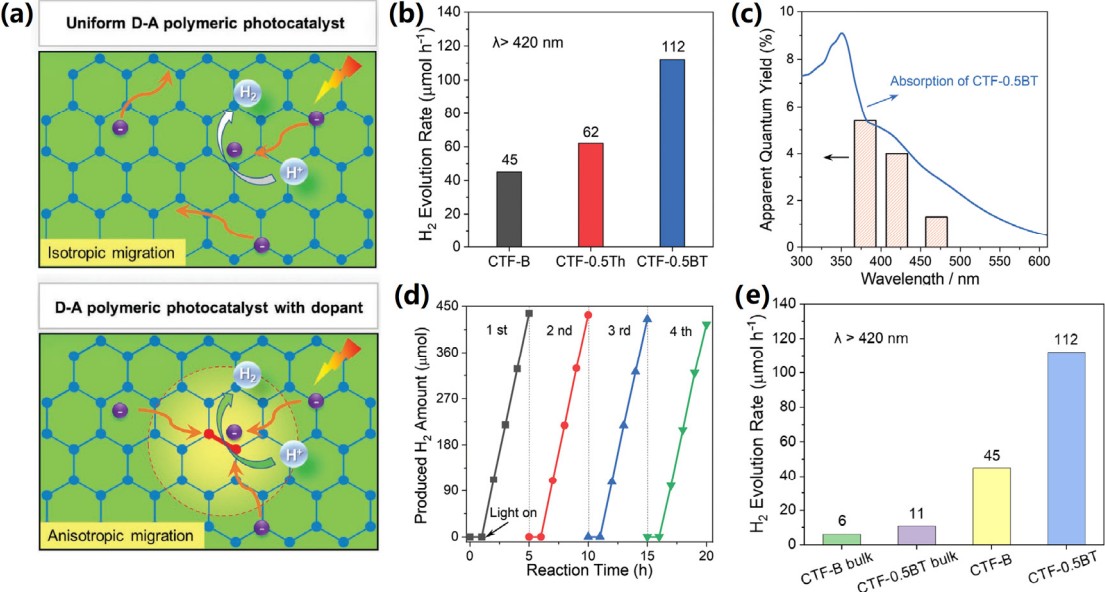

**Figure 10.** (**a**) Graphical representation of charge migration in two kinds of D–A polymer. (**b**) PHE rates of the polymers. (**c**) Wavelength dependence of AQY on PHE using CTF-0.5BT. (**d**) Time course of PHE of CTF-0.5BT. (**e**) The influence of morphology on the photocatalytic performance of CTFs [100].

Conjugated donor–acceptor polymers promote forward separation of intramolecular charges. However, the conjugated systems also suffer from backward charge recombination concurrently, which will result in lower photocatalytic activity and quantum efficiency. Therefore, it is of great significance to explore effective ways to inhibit charge recombination to improve photocatalytic efficiency. Guo and co-workers reported a strategy to promote charge separation and inhibit charge recombination by constructing a D–A$_1$–A$_2$ system in conjugated porous polymers (CPPs). CTFs (ter-CTF-X) with different ratios of donor (carbazole) and acceptors (triazine and benzothiadiazole) were constructed by polycondensation. The D–A$_1$–A$_2$ system utilizes the difference in the energy level gradients between different donors and acceptors to facilitate charge separation. Simultaneously, the copolymer constitutes an efficient photoinduced electron transfer system by electron transition between the HOMO and the LUMO, which allows electrons flow directionally from the donor units with higher energy levels to the acceptor units with lower energy levels, thus effectively retarding the backward recombination of charge (Figure 11a). As shown in Figure 11b,c, the PHE rate of the optimal terpolymer (ter-CTF-0.7) is up to 966 µmol h$^{-1}$ after adding Pt cocatalyst, which is 2 and 5.3 times higher than that of the original CTF–CBZ and CTF–BT, respectively. However, the PHE rate decreases gradually with the increase in

BT content. At the same time, the average PHE rate of ter-CTF-0.7 sample maintains great stability after five photocatalytic cycle tests (Figure 11d). In addition, ter-CTF-0.7 has the highest AQY at 420 nm, reaching a surprising 22.8%, which is superior to a majority of the CPPs, and also reaches 14.7% at 500 nm (Figure 11e). This work provides a new thought for the design and synthesis of photocatalytic systems based on CPPs to achieve maximum photocatalytic efficiency [101].

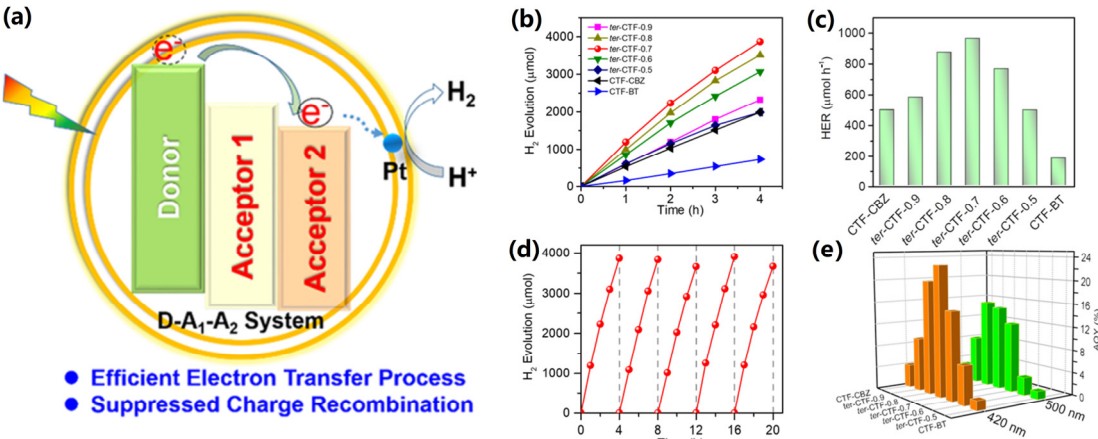

**Figure 11.** (**a**) D–$A_1$–$A_2$ system in the photoinduced electron-transfer process. (**b**) Time course of PHE of CTFs. (**c**) HER of CTFs. (**d**) Recycle tests for $H_2$ evolution. (**e**) AQY measurements of $H_2$ under 420 and 500 nm of CTFs [101].

Through CTF-based D–A, structure materials show higher PHE performance in a new way. However, the complicated preparation process is still an important research work in the future.

### 3.4. Other Photocatalysts

As is reported, the lifetime and mobility of the charge carriers in linear conjugated polymers are mainly determined by the intrinsic properties of the structural units, so the linear conjugated skeleton can be modified by some specific ways, such as introducing heteroatoms into the structural units and increasing the length and width of the conjugated chains [102–104].

In a recently constructed D–A architectonics work, Wang et al. designed a side-chain extended conjugated modification strategy to synthesize three polymers P0, P1, and P2 based on benzodithiophene(BDT) and dibenzothiophene S, S-dioxide(DBTSO). There are two additional thiophene units on the skeleton of P1 and P2 along the main-chain and side-chain directions, respectively. The conjugated functionalization of the side chain in P2 enhances the interchain π–π interaction, reduces the polymer bandgap, redshifts optical absorption bands, prolongs the exciton lifetime, regulates HOMO and LUMO energy levels, and improves the interchain transfer and transport of charges. Surprisingly, P2 reaches an PHE rate of 20 314 μmol $g^{-1}$ $h^{-1}$ in the presence of Pt cocatalyst and achieves a AQY 7.04% at 500 nm [105].

Tan et al. facilely synthesized a series of π-conjugated polymer photocatalysts based on EDOT through a strategy of atom-economic C-H direct arylation polymerization (DArP). SEM images exhibit that $BSO_2$-EDOT, DBT-EDOT, and DFB-EDOT show semblable layer-stacking morphology with flat surfaces, while Py-EDOT shows staggered rod-like structure (Figure 12a–d). The HER rate of D–A type $BSO_2$-EDOT is up to 0.95 mmol $h^{-1}$/6 mg, which is the highest among the linear CPs ever reported (Figure 12e). As shown in Figure 12f, the PHE rate of CPs dispersed in NMP is 1.5–44 times higher than that of CPs dispersed in MeOH, which is mainly attributed to the exfoliation effect in NMP that causes CPs to expose more active sites. Simultaneously, $BSO_2$-EDOT successfully achieves an apparent quantum yield of 13.6% at 550 nm (Figure 12g). In addition, in the presence of

NMP-based colloid, $BSO_2$-EDOT which is processed into a thin film and dispersed on a glass substrate still retains its photocatalytic activity (Figure 12h). The photocatalytic activities of CPs synthesized by DArP are significantly superior to those synthesized via Stile coupling, indicating that DArP is more beneficial to the preparation of CPs based on EDOT. The experimental results show that $BSO_2$-EDOT has enhanced hydrophilicity, enhanced electron-donating ability through C-O polar bonds and p-π conjugation, and improved electron transfer through D–A structure [106].

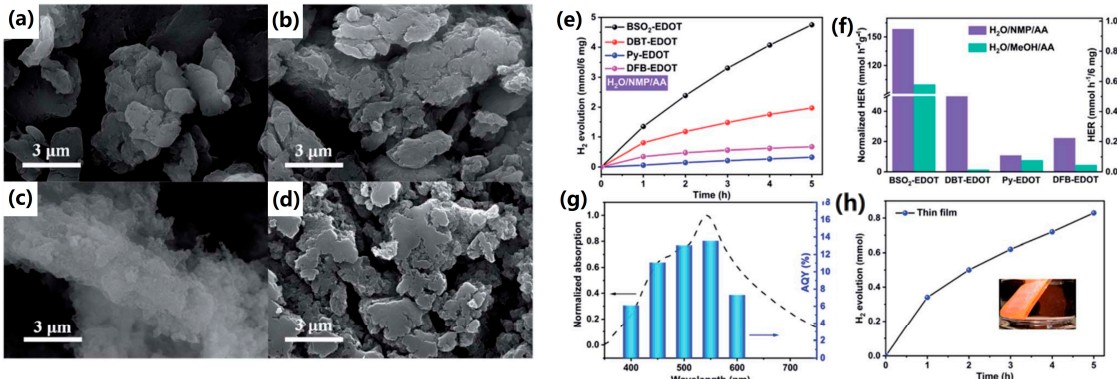

**Figure 12.** SEM images of (**a**) $BSO_2$–EDOT, (**b**) DBT–EDOT, (**c**) Py–EDOT, and (**d**) DFB–EDOT. (**e**) Photocatalytic hydrogen production as a function of time for 6 mg CPs under visible light irradiation. (**f**) Normalized PHE rate of CPs dispersed in AA/$H_2O$/NMP and AA/$H_2O$/MeOH. (**g**) AQYs of PHP for $BSO_2$–EDOT at five different incident light wavelengths. (**h**) PHP from $H_2O$ for the $BSO_2$–EDOT film (inset: film photograph) [106].

Although various organic semiconductor materials have been reported for constructing D–A structure photocatalyst for PHE application, it is still a grand challenge for the large-scale application to overcome the energy crisis. Other organic semiconductor materials are also worth further development and utilization to obtain efficient PHE performance in the future.

## 4. Conclusions and Outlook

In summary, this minireview describes D–A heterostructure photocatalyst for PHE application. We introduced that the unique D–A heterostructure is a system formed by direct connection of semiconductor components acting as donor and acceptor through covalent or non-covalent interaction, which is especially beneficial to realize the efficient separation and transfer of photogenerated charges under photoexcitation, so as to accelerate the PHE reaction. In addition, this review also summarizes the recent applications of various conjugated polymer-based D–A materials for PHE, including carbon-nitride-based polymers (g-$C_3N_4$), covalent organic frameworks (COFs), covalent triazine-based frameworks (CTFs), and other photocatalysts. Although some breakthrough progress has been made in the synthesis strategies and properties of these D–A-conjugated polymer-based photocatalysts, the PHE yield is still not sufficient to meet the industrial demand and there is much room for development. Based on D–A structure, enhancing visible light absorption, promoting the separation and transfer of charge carriers, inhibiting the rapid recombination of electron-hole pairs, and raising the number of active sites on the surface of the photocatalyst to improve the redox activity are the three key steps to optimize the photocatalytic performance of D–A photocatalyst. Inspired by these steps, this review presents some new analyses and perspectives based on the existing problems:

(i) D–A g-$C_3N_4$ is one of the most promising modification methods to improve the photocatalytic activity of g-$C_3N_4$ under visible light. In future works, further reasonable design of photocatalyst can be carried out in the following aspects. The separation and migration of photogenerated carriers can be further enhanced by loading suitable

cocatalyst onto the electron acceptor of D–A g-$C_3N_4$. In addition, the combination of molecular structure design and morphological modulation can also greatly improve the PHE performance, including the preparation of porous structures and ultra-thin nanosheet structures. Finally, the appropriate ratio of D–A monomer and the extension of conjugate structure also have significant effects on the photocatalytic activity.

(ii) The synthesis of highly crystalline D–A COFs requires many strict conditions, such as time, temperature, reaction rate, and preparation method. Therefore, the use of high throughput synthesis method is very promising. In addition, the active site of the reaction can be increased by anchoring uniformly dispersed metal atoms on D–A COFs. At the same time, the selection of appropriate skeleton and D–A repeating element are also extremely important to achieve the required geometric match between D and A elements.

(iii) Unlike COFs, which are linked by reversible covalent bonds, CTFs are composed of stable C=N bonds, which is not conducive to dynamic association and dissociation of bonds during the reaction. In addition, the synthesis of D–A CTFs generally requires relatively harsher reaction conditions, such as high temperature, high pressure, and strong acids or bases. Therefore, it is necessary to explore an environmental-friendly synthesis of CTFs. Finally, most CTFs are amorphous or have poor crystallinity, so some design strategies such as improving crystallinity or even achieving crystal plane modulation should be further investigated.

**Author Contributions:** Conceptualization, Y.Z. and J.W.; methodology, Y.Z.; software, J.S. and H.D.; validation, J.M. and Z.C.; formal analysis, X.Z. and C.L.; investigation, J.M. and Z.C.; resources, H.D. and Z.C.; data curation, Y.Z.; writing—original draft preparation, J.W. and Y.Z.; writing—review and editing, X.Z. and C.L.; visualization, J.S.; supervision, X.Z. and C.L.; project administration, H.D. and C.L.; funding acquisition, H.D. and C.L. All authors have read and agreed to the published version of the manuscript.

**Funding:** This work is supported by the National Natural Science Foundation of China (52072153), the Natural Science Foundation of Jiangsu Province (BK20190867), the Postdoctoral Science Foundation of China (2021M690023), Innovation and Entrepreneurship training Program for college students of Jilin Province (202210206038).

**Data Availability Statement:** The original contributions presented in the study are included in the article; further inquiries can be directed to the corresponding author.

**Conflicts of Interest:** The authors declare that there are no known competing financial interest or personal relationships in this paper.

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
