# Peer review of "Recent Progress in Conjugated Polymers-Based Donor–Acceptor Semiconductor Materials for Photocatalytic Hydrogen Evolution from Water Splitting"

_catalysts, doi:10.3390/catal13050850_

Round 1

Reviewer 1 Report

Thank you for the manuscript. It is well organized and publishable. Be consistent in abbreviations. Remember to use abbreviations after definition as long chemical names are taking time. Use scientific English.

Author Response

Thank you for the manuscript. It is well organized and publishable. Be consistent in abbreviations. Remember to use abbreviations after definition as long chemical names are taking time. Use scientific English.

Response: Thanks for these important suggestions. According to the suggestions, we further checked the abbreviations to make them consistent throughout the text. In addition, we also checked the whole manuscript based on and your marking manuscript, including some words, long difficult sentences, grammatical errors and the format of references.

Reviewer 2 Report

Check spaces when a reference is indicated e.g. 48

Indicate where the figures come from

Figure 2 is blurred

Figures in general contain several micrographs and graphs and are not well appreciated. 

In figure 9 which energy graph is shown?

Author Response

Comments and Suggestions for Authors

Check spaces when a reference is indicated e.g. 48.

Response: Thanks for this important suggestion. According to the suggestion, we have checked and revised all references in the revised manuscript.

Indicate where the figures come from.

Response: Thanks for this important suggestion. According to the suggestion, we have pointed out where Figures came from in the revised manuscript.

Figure 2 is blurred. Figures in general contain several micrographs and graphs and are not well appreciated. In figure 9 which energy graph is shown?

Response: Thanks for these important suggestions. According to the suggestions, we have polished all Figures for clarity and aesthetics in the revised manuscript.

Reviewer 3 Report

The review is of considerable interest and well done. I recommend it to be published after a minor revision.

1. The Authors should also proofread their manuscript (some spelling and grammar errors).

2. The author should better improve the beauty and quality of the figures in the manuscript.

3.Some publications are suggested to refer to improve the quality of the manuscript, such as: https://doi.org/10.3390/ma16062170

https://doi.org/10.3390/chemengineering7010004

https://doi.org/10.1016/j.ceramint.2022.05.151

https://doi.org/10.1007/s10904-023-02604-0

4. The conclusion is too short and also not targeted to the important aspects described in the review; please rephrase it.

5. Authors should write about the using of real wastewater in the manuscript and how the interaction occurred when there is different pollutants in it.

Hence, I recommend it accepted for publication after some minor revisions.

Author Response

Comments and Suggestions for Authors

The review is of considerable interest and well done. I recommend it to be published after a minor revision.

  1. The Authors should also proofread their manuscript (some spelling and grammar errors).

Response: Thanks for this important suggestion. According to the suggestion, we have revised manuscript, including some words, long difficult sentences, grammatical errors, and the format of references in the revised manuscript.

  1. The author should better improve the beauty and quality of the figures in the manuscript.

Response: Thanks for this important suggestion. According to the suggestion, we have polished all Figures for clarity and aesthetics in the revised manuscript.

3.Some publications are suggested to refer to improve the quality of the manuscript, such as: https://doi.org/10.3390/ma16062170

https://doi.org/10.3390/chemengineering7010004

https://doi.org/10.1016/j.ceramint.2022.05.151

https://doi.org/10.1007/s10904-023-02604-0

Response: Thanks for this important suggestion. We have consulted above works, which play an important guiding role for this manuscript. In addition, according to the suggestion, we have cited these important works in the revised manuscript.

  1. The conclusion is too short and also not targeted to the important aspects described in the review; please rephrase it.

Response: Thanks for these important suggestions. According to the suggestion, we have polished the conclusion. We hope the revised manuscript can meet the requirements of the catalysts.

  1. Authors should write about the using of real wastewater in the manuscript and how the interaction occurred when there is different pollutants in it. Hence, I recommend it accepted for publication after some minor revisions.

Response: Thanks for this important suggestion. The manuscript focused on the conjugated polymers-based donor−acceptor semiconductor materials for photocatalytic hydrogen evolution from water splitting. Therefore, we will discuss in the next work for the using of real wastewater and the interaction occurred among different pollutants.

Reviewer 4 Report

The mini-review is devoted to the analysis of works on conjugated polymeric materials for photocatalytic hydrogen evolution (PHE) during water splitting. I recommend including in the review the works devoted to conjugated microporous donor-acceptor type polymers. Such polymers exhibit excellent photocatalytic performance in water splitting due to their efficient isolation of light-induced excitons. For example, the work [Ahmed Fathi Saber, Ahmed M. Elewa, Ahmed F.M. EL-Mahdy. Donor to Acceptor Charge Transfer in Carbazole-based Conjugated Microporous Polymers for Enhanced Visible-Light-Driven Photocatalytic Water Splitting . ChemCatChem (2023). doi: 10.1002/cctc.202201287 ]. In addition, it would be useful to cite the work of [Sultan Otep, Tsuyoshi Michinobu, Qichun Zhang.  Pure Organic Semiconductor-Based Photoelectrodes for Water Splitting. Solar RRL (2020). doi: 10.1002/solr.201900395].

Author Response

Comments and Suggestions for Authors

The mini-review is devoted to the analysis of works on conjugated polymeric materials for photocatalytic hydrogen evolution (PHE) during water splitting. I recommend including in the review the works devoted to conjugated microporous donor-acceptor type polymers. Such polymers exhibit excellent photocatalytic performance in water splitting due to their efficient isolation of light-induced excitons. For example, the work [Ahmed Fathi Saber, Ahmed M. Elewa, Ahmed F.M. EL-Mahdy. Donor to Acceptor Charge Transfer in Carbazole-based Conjugated Microporous Polymers for Enhanced Visible-Light-Driven Photocatalytic Water Splitting . ChemCatChem (2023). doi: 10.1002/cctc.202201287 ]. In addition, it would be useful to cite the work of [Sultan Otep, Tsuyoshi Michinobu, Qichun Zhang.  Pure Organic Semiconductor-Based Photoelectrodes for Water Splitting. Solar RRL (2020). doi: 10.1002/solr.201900395].

Response: Thanks for this important suggestion. We have consulted above works, which play an important guiding role for this manuscript. In addition, according to the suggestion, we have cited these important works in the revised manuscript.